# Towards Algorithmic Diversity with Semantic Seed Sampling

## Abstract

Large language models (LLMs) combined with evolutionary search techniques have achieved remarkable results in challenging open-ended domains such as competitive programming and mathematical discovery. A key ingredient of such methods is solution space exploration, typically performed by sampling a large pool of candidates with high temperature. However, such sampling has been widely critiqued for providing little semantic diversity and introducing syntactic errors in structured domains such as code and math. We propose *semantic seed sampling*, a simple training-free method for controllable exploration. The model first generates a small set of semantically meaningful seeds (short hints or ideas), appends them to the task description, and samples solutions from each seed-conditioned prompt. We observe that semantic seed sampling explores disjoint neighborhoods of the solution space whose combined coverage is substantially larger than that of high-temperature sampling alone. As part of the Best-of-N pipeline, our method yields relative gains of up to 13.8%, while remaining token-efficient. We provide a theoretical explanation for the near-optimality of small per-seed budgets, supporting it with empirical evidence. These results highlight efficient solution space exploration as an underappreciated and promising direction for improving LLMs' problem-solving abilities.

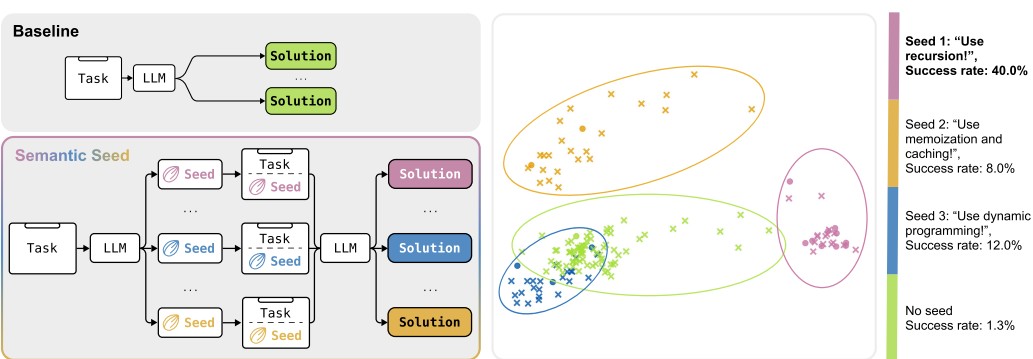

Figure 1: **SemSeed**: Semantic seed-conditioned sampling. Seeds target diverse solution space regions, allowing for controllable exploration under a fixed budget.

## 1 Introduction

The synergy of large language models (LLMs) and evolutionary search has recently delivered substantial progress in diverse open-ended areas such as robotics, neural architecture search, mathematics, algorithms, and code generation (Wu et al., 2024; Lange et al., 2024; Zhang et al., 2024). Their progress extends to benchmarks such as ARC-AGI, which has long been considered a litmus test for general reasoning capabilities (Chollet et al., 2024). The common pipeline consists of iterations of (1) generating a large pool of candidate solutions, (2) scoring with an external fitness function, such as unit tests, and (3) filtering and retaining the most promising ones (Fernando et al.; Li et al., 2022; Romera-Paredes et al., 2024; Novikov et al., 2025). Across all of the different approaches,

solid performance ultimately depends on the first sampling step, making the diversity and control of exploration a first-order concern.

A central challenge is how to induce *semantic* diversity under fixed inference budgets. The dominant approach is to raise the sampling temperature, but this is problematic for a number of reasons. First, higher temperature does not guarantee semantic diversity: since the solutions are sampled independently, with no specific control over which part of the solution space to explore, many generations may differ only in superficial phrasing (e.g., names of the functions, order of variables, etc.) while converging on the same underlying algorithm, wasting budget on redundant regions. Second, the effect of temperature is highly dependent on the task, the model, and the domain; in code generation in particular, higher temperatures often increase the rate of syntactic or structural errors, reducing the proportion of valid candidates (Shi et al., 2024a; Renze, 2024; Ding et al., 2023). Finally, recent work questions temperature as the right lever for creativity and shows that input-side randomness can help. For example, adding random prefix tokens during training and varying them at inference improves creative diversity for some minimal algorithmic tasks (Nagarajan et al., 2025). Despite the importance of diverse exploration, we still lack a simple and controllable method for steering generation toward distinct and complementary regions of the solution space.

We propose **semantic seed sampling** (SEMSEED), a training-free method that first elicits a small set of semantically distinct *seeds* (e.g., solution ideas, hints), appends each seed to the task to form modified prompts, and then distributes the sampling budget across these seed-conditioned prompts before applying standard Best-of-N selection. Intuitively, a semantic seed encodes a region of the solution space, responsible for a particular approach to the problem, like using recursion or dynamic programming (see fig. 1). Empirically, the seed-induced distribution consists of multiple disjoint modes, providing broader support than high-temperature baselines, while per-seed success rates are *sparse* – properties that enable provably efficient exploration under fixed budgets. On code benchmarks, this yields consistent improvements over strong baselines, with relative gains up to **13.8%**.

Our contributions:

- We introduce *semantic seed sampling*, a minimal change to Best-of-N sampling scheme that injects prompt-level diversity without additional training.
- We provide a mechanism study showing that seed-conditioned generations form separated modes with sparse success rates, explaining why breadth-wise exploration is effective; we also formalize conditions and some simple bounds connecting sparsity to pass@$k$ gains.
- We demonstrate consistent improvements on challenging code benchmarks (e.g., MBPP+, LiveCodeBench), and report ablations (temperature, number of seeds) that further support our theoretical assumptions.

## 2 RELATED WORK

A suite of code generation methods follows a sample-then-aggregate paradigm. AlphaCode demonstrated that large-scale Best-of-N sampling, combined with clustering and execution-based filtering, can solve competitive programming problems by surfacing diverse candidates and selecting those that pass tests (Li et al., 2022). Subsequent work systematizes this Best-of-N family: Self-Consistency draws multiple solutions (or reasoning traces) and aggregates to a final answer by majority voting, while execution-guided reranking/MBR frameworks (e.g., DOCE, CodeT) explicitly generate an $n$-best list and select using (self-generated) unit-test outcomes or minimum-risk criteria (Wang et al.; Li et al., 2024; Chen et al., a). Concurrently, hierarchical methods like CodeChain and FUNCODER instantiate the same idea at the component level: instead of sampling entire programs at once, they sample multiple candidates for subcomponents, use local selection (e.g., *functional consensus*), and then recompose a final solution, effectively performing Best-of-N within each node of a decomposition tree (Chen et al., 2024; Le et al.; Li et al., 2025). Another strain of methods, such as SELF-DEBUGGING, Reflexion, or MGDebugger, tries to iterate on the same program, employing the public unit test feedback to fix it (Shinn et al., 2023; Chen et al., b; Shi et al., 2024b).

Several recent works have explored ways to induce diversity in LLM outputs by varying or augmenting the prompt (or input) rather than relying solely on randomness in decoding or high-temperature sampling. (Naik et al., 2023) proposes DIV-SE and IDIV-SE, where the model is prompted to explore multiple reasoning approaches to the problem in a one-shot manner. Dipper (Lau et al., 2024)

introduces an LLM ensemble method where, first, a dataset-specific pool of prompt variations is generated. These prompt variations are then tested on a subset of examples and narrowed down to some pre-defined $n$ according to the accumulation of accuracy and diversity. Finally, the generation budget is distributed equally among the prompt versions, and the final result is aggregated by majority voting or with another LLM call. The most recent work (Nagarajan et al., 2025) introduces "*seed-conditioning*" by adding random prefixes to prompts as one way to inject diversity at the input side. The work demonstrates that seed-conditioning can, in certain cases, match or outperform output randomness (e.g., temperature sampling) in achieving diversity, particularly in minimalistic algorithmic scenarios.

# 3 METHOD

## 3.1 SETUP

We are given a textual description of the programming task $x$ and must produce a solution $y \sim p_\theta(y \mid x)$ (Python code) using an LLM with parameters $\theta$.

Let $c \colon X \times Y \to \{0, 1\}$ be the verifier function. It runs the solution code on all tests and checks whether all of them pass:

$$c(x, y) = \begin{cases} 1 & \text{if } y \text{ passes all public and private tests for task } x, \\ 0 & \text{otherwise.} \end{cases}$$

We define the per-task success rate $s(x)$ as the probability of generating a solution accepted by the verifier:

$$s(x) = \mathop{\mathbb{E}}_{y \sim p_\theta(y \mid x)} \big[ \mathbb{I}(c(x, y) = 1) \big].$$

Then, the pass@1 denotes the pass rate over the whole dataset (Kulal et al., 2019):

$$\text{pass@1} = \mathop{\mathbb{E}}_{x \sim D} \big[ s(x) \big].$$

If the samples $\{y_i\}_{i=1}$ are drawn i.i.d., then the probability of having at least one correct solution among $k$ samples is:

$$\text{pass@k} = \mathop{\mathbb{E}}_{x \sim D} \big[ 1 - (1 - s(x))^k \big].$$

In that case, there is an unbiased approximate to pass@k, expressed by the following formula (Chen et al., 2021):

$$\text{pass@k} = \mathop{\mathbb{E}}_{x \sim D} \left[ 1 - \frac{\binom{n-c}{k}}{\binom{n}{k}} \right] \qquad (0 \le k \le n),$$

where $n$ is the total number of generated solutions for task $x$, and $c$ is the number of correct ones.

## 3.2 SEMSEED($K@N$): SEED-CONDITIONED BEST-OF-N

Our method mirrors Best-of-N in the selection stage but changes how candidates are sampled by adding a preceding *semantic seed* generation step. The aim of it is to diversify solutions and improve coverage.

1. *Seed generation.* We start by obtaining $K$ seeds:

$$h = (h_1, \ldots, h_K) \sim p_\theta(h \mid x)$$

   Practically, we prompt the model to generate a list of $K$ short, diverse solution ideas or *hints*. Note that all $h_i$ are generated at once, so they are not independent. We prefer sampling at once instead of i.i.d. sampling to promote diversity, since the LLM will unlikely generate similar seeds in the same list.

2. *Modified tasks.* For each seed, we form a modified task:

$$x_i = [x, h_i], \qquad i = 1, \ldots, K$$

   In practice, this can be done by concatenating with a specific prompt form, for example *"Use the hint below: {hint}"*.

3. *Seed–conditioned sampling.* We fix a total budget of $N$ solutions, and distribute the sampling budget across the $K$ modified task examples $\{x_i\}_{i=1}^K$.

Let $m : \{1, \ldots, N\} \to \{1, \ldots, K\}$ map each sample index $j$ to the seed it uses; equivalently, $n_i = \big|\{j : m(j) = i\}\big|$ is the number of samples obtained from the modified task $x_i$, with $\sum_{i=1}^K n_i = N$. We then draw

$$y_j \sim p_\theta\big(y \mid x_{m(j)}\big), \qquad j = 1, \ldots, N$$

By default, we use a near-uniform allocation $n_i = \lfloor N/K \rfloor$ and distribute the remainder arbitrarily. Under uniform allocation, the overall sampling distribution is the mixture:

$$\frac{1}{K} \sum_{i=1}^K p_\theta(y \mid x_i)$$

4. *(Optional) Selection (as in Best-of-N).* We rank all candidates by the number of passed public tests and return the top one, as in Best-of-$N$, if one solution is required.

Unlike Best-of-$N$, which samples all $y_i \sim p_\theta(y \mid x)$ independently from the same distribution, SEMSEED samples from seed-conditioned distributions $p_\theta(y \mid x_{m(j)})$ induced by LLM-generated seeds $h_i$. This design encourages coverage across distinct solution strategies and enables a form of *controllable exploration*: each seed defines a targeted region of the solution space, and the sampling budget is explicitly divided across these regions rather than concentrated in a single mode. Because the seeds are diverse, the resulting coverage is broader and more balanced. Moreover, seeds can also be externally specified (e.g., by domain experts or larger LLMs) to direct the search toward particular strategies or areas of interest.

## 3.3 BOUNDS ON PASS@$k$

First, let's define the success rate for the seed-induced distribution:

$$s_{h_i}(x) = \mathop{\mathbb{E}}_{y \sim p_\theta(y \mid [x, h_i])} \big[\mathbb{I}\left(c(x, y) = 1\right)\big]$$

Seeds $h_i$, in general, are not independent, so we cannot immediately use the same formula as Chen et al. (2021) for estimating pass@$k$. However, we can derive an upper and lower bounds for it:

**Prop. 1** (Bounds on pass@$k$ for seed-conditioned sampling). *Let $x$ be a task, and $p_\theta(y \mid x)$ be the distribution over the solution space, parameterized by the LLM with weights $\theta$. Let the seeds be $h = (h_1, \ldots, h_K)$ with corresponding success rates $s_{h_i}(x)$, where $K \leq k$, sampling $n_i$ solutions for seed $h_i$, $\sum_{i=1}^K n_i = k$. Then the probability pass@$(k, x, h)$ of at least one sample being correct among $k$ generated for the task $x$ with seeds $h$:*

$$\min\left(1, K - \sum_{i=1}^K (1 - s_{h_i}(x))^{n_i}\right) \geq pass@(k, x, h) \geq 1 - \min_{i \in \{1, \ldots, K\}} (1 - s_{h_i}(x))^{n_i}.$$

For the proof, please refer to appendix A.1.

The Proposition 1 tells us that under the uniform per-seed sampling budget $n_i = n$, we have:

$$\text{pass@}(k, x, h) \geq 1 - (1 - s_{h^\star}(x))^n$$
$$s_{h^\star}(x) = \max_{i \in \{1, \ldots, K\}} s_{h_i}(x)$$

This has a simple implication. If the per-seed success rates are sparse, close to either zero or one, then the failure probability $(1 - s_{h_i}(x))^n$ collapses to zero either very quickly (good seed) or extremely slowly (bad seed). In other words, once a **winning seed** $s_{h^\star}(x) \gg s(x)$, an idea that will

Table 1: Results on HumanEval+ and MBPP+. pass@1 is shown in %.

| Model name | Sampling method | HumanEval+ | MBPP+ |
|---|---|---|---|
| DeepSeek-Coder-V2-Lite | Standard | 75.9 | 59.4 |
| | CoT | 75.3 | 58.3 |
| | Best-of-6 | 78.4 | 62.8 |
| | SELF-DEBUGGING@6 | 77.8 | 66.3 |
| | FUNCODER* | 36.4 | 35.3 |
| | SEMSEED(6@6) | **85.2** | **68.4** |
| Qwen2.5-Coder-14B | Standard | 84.0 | 60.7 |
| | CoT | **87.0** | 63.6 |
| | Best-of-6 | 86.4 | 69.0 |
| | SELF-DEBUGGING@6 | **87.0** | 71.9 |
| | FUNCODER* | 39.5 | 42.2 |
| | SEMSEED(6@6) | 85.1 | **73.0** |
| Qwen2.5-Coder-7B | Standard | 78.4 | 46.3 |
| | CoT | 80.9 | 51.9 |
| | Best-of-6 | **87.0** | 65.5 |
| | SELF-DEBUGGING@6 | 84.6 | 62.3 |
| | FUNCODER* | 32.1 | 41.2 |
| | SEMSEED(6@6) | 85.2 | **67.1** |

likely lead to a correct solution, is found, only a handful of samples are typically needed to obtain a correct solution with high probability.

*Under this sparsity assumption, it is therefore more efficient to spend the budget on **searching for good seeds**, exploring new regions of the solution space, rather than heavily exploiting a single seed in the hope of eventually striking success.*

## 4 EVALUATION RESULTS

### 4.1 SETUP

In this section, we evaluate three open-source code-tuned LLMs on four commonly used code generation benchmarks. Concrete details of the sampling parameters and benchmark statistics can be found in appendix A.2. Among simple baselines like Chain-of-Thought and Best-of-N, we use code-specific ones like SELF-DEBUGGING and FUNCODER. For a complete list with descriptions, please refer to appendix A.3.

### 4.2 CODE GENERATION RESULTS

The results are presented in tables 1 and 2. Across models, Best-of-N is the strongest single-prompt baseline. Our SEMSEED matches or exceeds the best baseline on all benchmarks, with the largest gains on MBPP+ and LiveCodeBench. Concretely, we outperform all baselines on MBPP+ and LiveCodeBench for every model, and we are on par with the best baseline on HumanEval+ and xCodeEval. The largest relative improvement appears on LiveCodeBench with DeepSeek-Coder-V2-Lite: +3.0 absolute (+13.8% relative) over BoN.

On MBPP+ and LiveCodeBench, many tasks admit multiple algorithmic strategies (different decompositions, data-structure choices, or edge-case policies). BoN at high temperature tends to revisit similar neighborhoods; SEMSEED explicitly opens multiple "idea-conditioned" niches and spreads the budget across them, which increases the odds of touching a high-value mode. This aligns with our mechanism study: seed-conditioned distributions are separated, and the baseline misses strong modes surprisingly often. On HumanEval+, scores are already saturated by dense public tests: both BoN and SEMSEED routinely achieve 94–96% public success before final selection. In this regime,

Table 2: Results on xCodeEval and LiveCodeBench. pass@1 is shown in %.

| Model name | Sampling method | xCodeEval | LiveCodeBench |
|---|---|---|---|
| DeepSeek-Coder-V2-Lite | Standard | 23.6 | 13.8 |
| | CoT | 23.2 | 13.8 |
| | Best-of-6 | **31.0** | 21.6 |
| | SELF-DEBUGGING@6 | 26.4 | 18.6 |
| | FUNCODER* | 18.6 | 13.2 |
| | SEMSEED(6@6) | 30.8 | **24.6** |
| Qwen2.5-Coder-14B | Standard | 21.4 | 24.0 |
| | CoT | 18.4 | 22.8 |
| | Best-of-6 | **29.6** | 32.3 |
| | SELF-DEBUGGING@6 | 23.2 | 27.5 |
| | FUNCODER* | 19.7 | 23.9 |
| | SEMSEED(6@6) | 28.6 | **34.1** |
| Qwen2.5-Coder-7B | Standard | 12.8 | 16.2 |
| | CoT | 12.6 | 15.6 |
| | Best-of-6 | **21.4** | 26.3 |
| | SELF-DEBUGGING@6 | 18.0 | 23.4 |
| | FUNCODER* | 12.0 | 15.0 |
| | SEMSEED(6@6) | 21.0 | **26.9** |

additional exploration provides little headroom: improvements would mostly come from better selection or solution refinement rather than broader search.

For FUNCODER, we observed relatively high variance across the three runs. In our experiments, we followed the authors' implementation closely, using the same prompts as well as similar parameters for breadth and depth control and for temperature settings across stages, while adapting the method to LiveCodeBench. We hypothesize that the performance of FUNCODER may be sensitive to the combination of the model and the decoding parameters, which could explain the variance observed and the difference in scores from the original article.

In general, semantically guided exploration consistently helps where the solution landscape is structured into multiple distinct modes and baseline sampling is redundant (MBPP+, LiveCodeBench), and it remains competitive when test density or canonical structure limits the upside of diversity (HumanEval+, xCodeEval).

### 4.3 EXPLORATION ANALYSIS

In this subsection, we attempt to analyze how semantic seeds change the solution space exploration process to understand why the method works and how it differs from the Best-of-N baseline.

Table 3: Solution space analysis scores. ↑ means higher is better

| Metric | Value |
|---|---|
| Modes separation (ARI, ↑) | 0.49 |
| Modes separation (Acc., ↑) | 67% |
| Baseline-to-seed coverage (↓) | 55% |
| Seed-to-baseline coverage (↑) | 77% |
| Best mode miss rate (↓) | 41% |

**Setup.** We form a small analysis dataset as a subset of MBPP+, employ DeepSeek-Coder-V2-Lite to sample solutions, and obtain solution embeddings using a Sentence Transformer (Reimers & Gurevych, 2019). For further details, please refer to appendix A.2.

**Seeds encode distinct regions of solution space.** First, we test whether the seeds steer generation into distinct regions of the solution space, providing control over the exploration process. In other words, we investigate whether they share support or not:

$$\sup\{p_\theta(y \mid x_i)\} \cap \sup\{p_\theta(y \mid x_j)\} = \varnothing, \; i \neq j$$

To test this, for each sample, we cluster solutions generated from different seeds and measure how well the clusters align with the true seed IDs. Specifically, for each task description we generate K=5 seeds, sample 20 solutions per seed, and repeat the process three times, averaging across task samples and seed sets. We then apply K-Means (Lloyd, 1982) with K=5, and evaluate clustering quality using the Adjusted Rand Index (ARI) (Rand, 1971) and clustering accuracy (Hungarian-matched). The Adjusted Rand Index (ARI) measures how well two clusterings agree by counting pairwise co-assignments, then correcting for chance, where 0 indicates random agreement and 1 indicates perfect alignment. The results, reported in table 3, show an average ARI of 0.49 and accuracy of 67% (chance = 20%), indicating that seed-conditioned generations carve out distinct and narrow basins of solution space, rather than merely rephrasing the same local neighborhood, and thus explore different solution strategies. Please, see fig. 4 for examples visualization, and table 6 for metrics explanation.

**Seed-conditioned sampling is more diverse.** Here, we examine whether seed conditioning increases the overall diversity of the generations. By diversity here we mean the effective support of the distribution: if the seed-induced distribution covers more regions of the solution space, exploration under the same budget is more likely to encounter distinct strategies. To test this, we use previously obtained K-Means ($K = 5$) for seed-conditioned generations, and compare their coverage with baseline samples along two directions.

First, we ask: *how many seed-induced modes are reached by the baseline?* Practically, we want to measure the following:

$$\sup\left\{\frac{1}{K}\sum_{i=1}^{K} p_\theta(y \mid x_i)\right\} \setminus \sup\{p_\theta(y \mid x)\}$$

We assign each baseline sample to its nearest seed cluster and count the fraction of seed clusters that contain at least one baseline solution.

Second, we ask: *how much of the baseline support is covered by the seeds?* We approximate seed support as the union of balls $\varepsilon_i$ centered at seed cluster centroids $c_i$ with radius $r_i$ equal to the 95-th percentile of intra-cluster distances, and measure the fraction of baseline samples lying inside this union:

$$\sup\left\{\frac{1}{K}\sum_{i=1}^{K} p_\theta(y \mid x_i)\right\} \approx \bigcup_{i=1}^{K} \varepsilon_i, \quad \varepsilon_i = \{z \mid ||z - c_i||_2^2 \leq r_i\}$$

The results show that baseline generations reach only 55% of the seed clusters on average (roughly 2–3 out of 5), indicating that large parts of the solution space remain unexplored without seeds. In contrast, we find that 77% of baseline solutions lie inside the seed support, confirming that the seed-induced distribution not only practically covers the baseline but also expands well beyond it. This demonstrates that seed conditioning provides broader support and thus greater diversity, enabling exploration of solution space regions that high-temperature sampling alone fails to reach.

**Seed-conditioned exploration is more efficient.** Finally, we analyze how seeds affect the efficiency of exploration under a fixed budget. Our analysis in section 3.3 predicts that if per-seed success rates are sparse, then only a small budget is required to exploit a good seed, making breadth-wise exploration across multiple seeds more efficient than oversampling a single one. Empirically, this assumption is supported: per-seed success rates are indeed sparse (see fig. 2), with most seeds either failing completely or succeeding reliably.

Moreover, the seeds that succeed are highly valuable. On average, the best seed $h^\star$ achieves a success rate that is 21% higher than the baseline distribution:

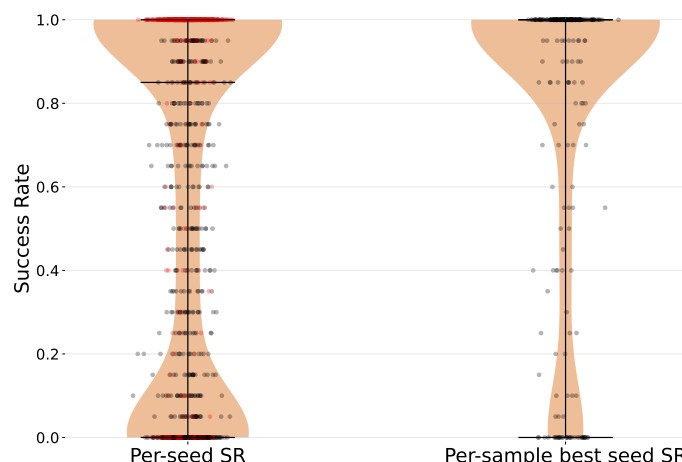

Figure 2: **Left**: per-seed success rate distribution $s_{h_i}(x)$. The red dot $\cdot$ signifies the best seed $h^\star$ in the generation $(h_1, \ldots, h_K)$. **Right**: per-sample best seed success rate distribution $s_{h^\star}(x)$. The success rate of seed-based generations $y \sim p_\theta(y \mid x_i)$ is bimodal and sparse.

Table 4: Effect of number of seeds ($K$) on pass@1.

| Method | HumanEval+ | MBPP+ | xCodeEval | LiveCodeBench |
|---|---|---|---|---|
| SEMSEED(2@6) | 82.1 | 66.0 | 30.0 | 22.2 |
| SEMSEED(3@6) | 82.7 | 67.6 | **31.2** | 23.3 |
| SEMSEED(6@6) | **85.2** | **68.4** | 30.8 | **24.6** |

$$s_{\text{diff}} = \mathop{\mathbb{E}}_{\substack{x \sim D \\ h \sim p_\theta(h|x)}} [s_{h^\star}(x) - s(x)] > 0.21$$

verified by the Wilcoxon signed rank test ($p < 0.05$).

In fact, the baseline fails to cover these best seeds in 41% of tasks, underscoring that it not only undersamples the distribution but also frequently misses the most promising solution space areas. Together, these findings confirm that seed-conditioned sampling enables more efficient use of budget by prioritizing exploration of new regions over repeated sampling from the same distribution.

### 4.4 ABLATIONS

Here, we run a few ablations with the DeepSeek-Coder-V2-Lit model. The setup is similar to the previous section, more details can be found in appendix A.2.

**Number of seeds $K$.** We vary the number of seeds while preserving the total sampling budget fixed (see table 4). Increasing $K$ from $2 \to 6$ consistently helps on HumanEval+ ($82.1 \to 85.2$) and MBPP+ ($66.0 \to 68.4$), and yields the best LiveCodeBench score at $K = 6$ ($22.2 \to 24.6$). xCodeEval peaks at $K = 3$ ($31.2$) and stays close at $K = 6$ ($30.8$). These trends support our assumptions on the exploration mechanism: seeds induce narrow, distinct modes with sparse success, so breadth-wise exploration increases the chance that at least one high-yield region is included, and only a modest per-seed depth is needed once such a mode is present.

**High temperature.** We ablate on DeepSeek-Coder-V2-Lite by raising the temperature from default 0.3 to 1.0 (see table 5). For SEMSEED, we denote $T_s$ for seed generation step temperature, and $T_g$ for solution sampling. BoN benefits from higher $T$ on three of four datasets (HumanEval+ $78.4 \to 83.3$, MBPP+ $62.8 \to 68.7$, LiveCodeBench $21.6 \to 23.3$; xCodeEval unchanged at $31.0$). With higher seed temperature $T_s$, SEMSEED improves on the harder suites (MBPP+ $68.4 \to 70.6$, xCodeEval $30.8 \to 31.6$, LiveCodeBench $24.6 \to 25.7$) and dips on HumanEval+ ($85.2 \to 82.7$).

Table 5: High temperature effects on pass@1. For SEMSEED, $T_s$ denotes the temperature at the seed generation step, and $T_g$ at the solution generation.

| Temperature | Method | HumanEval+ | MBPP+ | xCodeEval | LiveCodeBench |
|---|---|---|---|---|---|
| $T = 0.3$ | Best-of-6 | 78.4 | 62.8 | 31.0 | 21.6 |
| $T = 1.0$ | Best-of-6 | 83.3 | 68.7 | 31.0 | 23.3 |
| $T_s = T_g = 0.3$ | SEMSEED(6@6) | **85.2** | 68.4 | 30.8 | 24.6 |
| $T_s = 1.0,\ T_g = 0.3$ | SEMSEED(6@6) | 82.7 | 70.6 | **31.6** | **25.7** |
| $T_s = T_g = 1.0$ | SEMSEED(6@6) | 84.0 | **71.4** | **31.6** | **25.7** |

With both $T_s$ and $T_g$ high, SEMSEED gains benefits from both structured exploration and inter-cluster diversity. Overall, SEMSEED remains best or tied on the more challenging benchmarks and competitive elsewhere, indicating that controlled exploration is beneficial in both low and high temperature settings.

**Token efficiency**   In terms of token usage, Standard and Chain-of-Thought prompting are the most economical, consuming roughly N times fewer tokens than Best-of-N, since they produce only a single solution per task. SELF-DEBUGGING@N is somewhat more token-efficient than BoN, though the magnitude of the savings varies across models. Our method is nearly as token-efficient as BoN, with differences becoming negligible on datasets that require longer solutions (e.g., LiveCodeBench, xCodeEval). The least token-efficient approach is FUNCODER, whose hierarchical decomposition and functional consensus introduce substantial oversampling at the component level. For complete picture, please see fig. 3 in appendix A.2.

## 5   LIMITATIONS

While our approach provides consistent gains, the imperfect clustering scores (e.g., ARI below 1.0) indicate that seeds do not yet achieve complete separation. Though rare in practice, occasional misalignment of the model may result in neglecting the seed, especially at higher temperatures. The complete coverage of valuable areas is not guaranteed, as suggested by suboptimal coverage scores, and the granularity of exploration, i.e., the effective size of clusters, is determined only implicitly by how specific or general the seed formulations are.

## 6   CONCLUSION

We introduced semantic seed sampling, a simple inference-time method for diversifying exploration in LLM-based search. Our experiments show consistent gains of up to 13.8%, enabled by exploration control through support separation across seeds, and higher diversity, secured by broader coverage of the solution space compared to the temperature sampling. These results underscore the importance of exploring beyond naive sampling, and position our work as an early step toward more efficient and structured approaches to exploration for advanced code generation and reasoning.

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

# A  APPENDIX

## A.1  PROPOSITION

Here, we prove the main theoretical results of the paper.

**Prop. 2** (Bounds on pass@k for seed-conditioned sampling)**.** *Let $x$ be a task, and $p_\theta(y \mid x)$ be the distribution over the solution space, parameterized by the LLM with weights $\theta$. Let the seeds be $h = (h_1, \ldots, h_K)$ with corresponding success rates $s_{h_i}(x)$, where $K \leq k$. Then the probability pass@$(k, x, h)$ of at least one sample being correct among $k$ generated for the task $x$ with seeds $h$:*

$$\min\left(1, K - \sum_{i=1}^{K}(1 - s_{h_i}(x))^{n_i}\right) \geq pass@(k, x, h) \geq 1 - \min_{i \in \{1, \ldots, K\}}(1 - s_{h_i}(x))^{n_i}.$$

*Proof.* Let $A_i = \prod_{i=1}^{n_i} \mathbb{I}[c(x, y_i) = 0]$. Then $\{A_i = 1\}$ corresponds to the event of "all $n_i$ solutions generated with seed $i$ are wrong". Since per-seed samples are i.i.d., we have:

$$\mathbb{P}(A_i = 1) = \mathbb{P}(\prod_{i=1}^{n_i} \mathbb{I}[c(x, y_i) = 0] = 1) = \mathbb{P}(c(x, y_1) = 0, \ldots c(x, y_{n_i}) = 0) = (1 - s_{h_i}(x))^{n_i}.$$

Then, the pass@$(k, x, h)$ is defined as the probability of the event when "at least one correct sample in $k = \sum_{i=1}^{K} n_i$ generations for task $x$ with seeds $h$":

$$\text{pass@}(k, x, h) = 1 - \mathbb{P}\left(A_1 = 1, \ldots, A_K = 1\right)$$

Using Fréchet inequalities (Billingsley, 2017) for events $\{A_i = 1\}_{i=1}^{K}$:

$$\max\left(0, \sum_{i=1}^{K} \mathbb{P}(A_i = 1) - (K - 1)\right) \leq \mathbb{P}\left(A_1 = 1, \ldots, A_K = 1\right) \leq \min_{i \in \{1, \ldots, K\}} \mathbb{P}(A_i = 1)$$

where

$$\min_{i \in \{1, \ldots, K\}} \mathbb{P}(A_i = 1) = \min_{i \in \{1, \ldots, K\}}(1 - s_{h_i}(x))^{n_i}$$

and so, for the lower bound we have:

$$\text{pass@}(k, x, h) \geq 1 - \min_{i \in \{1, \ldots, K\}}(1 - s_{h_i}(x))^{n_i}.$$

and for the upper bound:

$$\max\left(0, \sum_{i=1}^{K} \mathbb{P}(A_i = 1) - (K - 1)\right) = \max\left(0, \sum_{i=1}^{K}(1 - s_{h_i}(x))^{n_i} - (K - 1)\right)$$

$$1 - \max\left(0, \sum_{i=1}^{K}(1 - s_{h_i}(x))^{n_i} - (K - 1)\right) = \min\left(1, K - \sum_{i=1}^{K}(1 - s_{h_i}(x))^{n_i}\right)$$

$$\text{pass@}(k, x, h) \leq \min\left(1, K - \sum_{i=1}^{K}(1 - s_{h_i}(x))^{n_i}\right)$$

$\square$

## A.2 Evaluation results and analysis

In general, we evaluate three open LLMs: DeepSeek-Coder-V2-Lite (Zhu et al., 2024), Qwen2.5-Coder-7B, and Qwen2.5-Coder-14B (Hui et al., 2024), across four code generation benchmarks. HumanEval+ and MBPP+ serve as the easier, well-studied suites with dense test coverage; xCodeEval and LiveCodeBench are more challenging (see table 8). We use nucleus sampling with each model's recommended default decoding hyperparameters (see table 7). Following FUNCODER, we run three independent trials for every method and report the median score, unless specified otherwise.

**Analysis setup.** To form the analysis dataset, we select 100 random tasks from MBPP+. We employ DeepSeek-Coder-V2-Lite (instruction-tuned) to generate solutions. For the baseline, we sample 100 solutions per task under high temperature ($T = 1.0$, while the default is $T = 0.3$). For seed-conditioned sampling, we generate $K = 5$ seeds per problem with high temperature ($T = 1.0$) and $n_i = 20$ solutions per seed with default low temperature, totaling to 100 samples per seed. We average the metrics over 3 seed-set generations to account for seed variability. To obtain embeddings of the solutions, we use the BGE Large v1.5 sentence encoder for the English language (Xiao et al., 2023). For a short explanation of the clustering metrics used, please refer to table 6.

To evaluate how well seed-conditioned generations separate into distinct regions, we report two clustering metrics. **The Adjusted Rand Index (ARI)** (Rand, 1971) measures the agreement between predicted clusters and ground-truth seed IDs, correcting for chance: ARI = 0 corresponds to random clustering and ARI = 1 to perfect alignment. It works by looking at all pairs of points and checking whether each pair is assigned together or apart in both clusterings. The raw agreement is then adjusted by subtracting the expected agreement under random clustering. **The clustering accuracy** is computed by optimally aligning predicted clusters with seed IDs using the Hungarian matching algorithm (Kuhn, 1955) and then measuring the fraction of correctly assigned samples; chance level is 1/K (e.g., 20% for $K = 5$). Together, these metrics quantify how consistently generations from the same seed form distinct clusters.

**Ablations setup.** Similarly to the analysis setup, we only run DeepSeek-Coder-V2-Lite model. We report the median results across 3 runs, varying temperature and other parameters.

Table 6: Metrics explained. The ↑ means higher is better.

| Metric | Range | Random | Intuition |
|---|---|---|---|
| Adjusted Rand Index ↑ | $[0, 1]$ | 0 | How well the clustering matches the true seed groups, after correcting for chance overlaps |
| Clustering accuracy ↑ | $[0, 1]$ | $1/K$ | The fraction of correctly grouped samples after optimally matching predicted clusters to true seed labels |

**Model sampling parameters.** Defaults are taken from each model's `generation_config.json` on Hugging Face. DeepSeek-Coder-V2-Lite is a Mixture-of-Experts model, so "active" parameters indicate the number of parameters used per token during inference. All the models are used in instruction-tuned versions. We leave out the "-Instruct" to save space. We perform nucleus sampling (Holtzman et al.) with default parameters, specified in table 7.

Table 7: Models used and their default decoding settings. For MoE models, we report total parameters and (active) parameters used per token.

| Model | # Params | Temperature | Top p |
|---|---|---|---|
| DeepSeek−Coder−V2−Lite | 16B (2.4B active) | 0.3 | 0.95 |
| Qwen2.5−Coder−7B | 7.6B | 0.7 | 0.80 |
| Qwen2.5−Coder−14B | 14.7B | 0.7 | 0.80 |

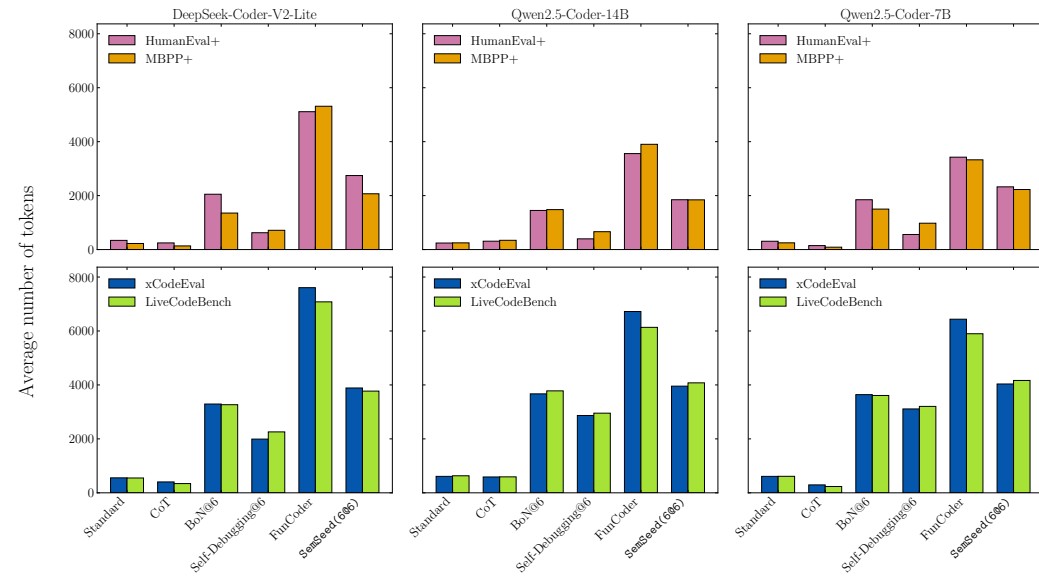

Figure 3: The average token consumption across datasets and methods. Here, only generated tokens are counted.

**Datasets.** We used EvalPlus (Liu et al., 2023) versions of HumanEval and MBPP datasets, as they contain significantly more tests, which allows for rigorous evaluation. For LiveCodeBench (Jain et al.), in order to reduce contamination risk, we use only release v5, yielding 167 problems published after September 2024. We also evaluate on xCodeEval (Khan et al., 2023) using the same split reported by FUNCODER. Across all datasets, we limit the number of public test cases to 3.

Table 8: Dataset statistics for code generation benchmarks used in this work.

| Dataset | # Samples | Avg. tests / sample | Split / Notes |
|---|---|---|---|
| HumanEval+ | 164 | $\approx 760$ | EvalPlus augmentation (80× tests) |
| MBPP+ | 378 | $\approx 108$ | EvalPlus v0.2.0 (trimmed $399 \rightarrow 378$) |
| xCodeEval | 500 | $\approx 60$ | Same split as used by FUNCODER |
| LiveCodeBench | 167 | $\approx 39$ | v5 time-span subset (contamination-minimized) |

## A.3 BASELINES

**Standard & Chain-of-Thought.** As the most straightforward baseline, we prompt the LLM to produce the Python code solution directly (Standard). The Chain-of-Thought (COT) (Wei et al., 2022) differs only in the addition of the step-by-step analysis request phrase.

**SELF-DEBUGGING@$N$.** Given the task, we run $N$ iterations of SELF-DEBUGGING (unit test feedback variation) (Chen et al., b), using public test cases. We stop and return the code if it passes all public test cases, or if we reach the maximum number of self-debugging iterations $N$.

**FUNCODER.** This is a recent state-of-the-art hierarchical code-generation method that dynamically decomposes a task into sub-functions using a divide-and-conquer strategy, then recomposes solutions bottom-up (Chen et al., 2024). To sample correct sub-functions and avoid upward error propagation, it employs *functional consensus* by sampling a pool of candidates, measuring their functional similarity, and returning the one with the highest aggregated score.

**Best-of-N.** In the standard Best-of-N baseline we draw $N$ i.i.d. candidates from the same conditional distribution:

$$y_i \sim p_\theta(y \mid x), \qquad i = 1, \ldots, N$$

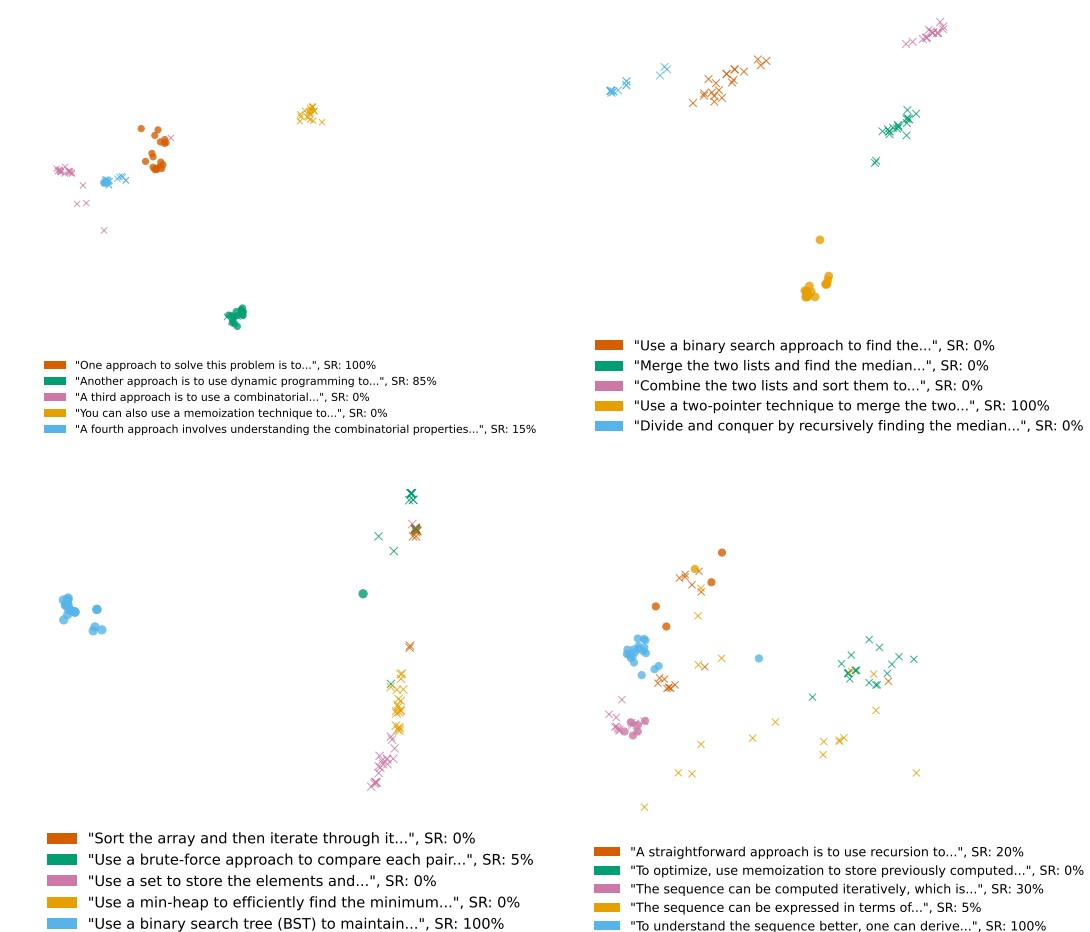

Figure 4: Examples of seed-conditioned generated solutions. The colors correspond to the specific seed, and the circle dots correspond to the correct solution. To visualize, we perform PCA on the embeddings.

We then execute the public tests for each candidate and select the winner:

$$y_w = \arg \max_{i \in \{1, \ldots, N\}} \#\text{passed}(y_i)$$

which is used to measure pass@1 on the private test suite.

For every baseline, we release the prompts in appendix A.4.

## A.4 Prompts

## A.5 Use of Large Language Models

Large language models were used solely for paraphrasing and polishing the text of the manuscript. There was no involvement in the research design, experimentation, analysis, or generation of results.

**Ethics statement.** All experiments rely on publicly available datasets used in accordance with their original licenses. No new human data or personal annotations were collected.

**Reproducibility statement.** We use only publicly available models, datasets, and baselines. Comprehensive details on splits, optimization, and evaluation protocols are included in the main text, with further specifics in the appendix. The full code base will be released to support replication. Experiments were executed on distributed setups of 2–4 nodes, each equipped with 4 AMD MI250x GPUs.

**System Prompt**

```
You are a helpful AI assistant, specializing in solving coding problems in Python.
```

**Seed Prompt**

```
Please, provide {n_seeds} solution ideas for the following problem:

{task}

Make sure that your idea / hint are helpful, full, and diverse. Keep each idea a one line.
This is the time to explore different approaches to the problem.
Enclose your response in ```json``` at the end of your answer, e.g.:
```json
[
    "Idea 1",
    "Idea 2",
    "Idea 3"
]
```
```

**Solution Prompt**

```
Solve the following coding problem. Write code in Python. Encolse your code in a code block
(```python...```).

{task}

Here is a hint for you, please use it to solve the problem:

{seed}
```

Figure 5: SEMSEED prompts.

**Solution Prompt**

```
Solve the following coding problem. Write code in Python. Encolse your code in a code block
(```python...```).

{task}
```

Figure 6: Solution prompt used by Standard and Best-of-N baselines. The system prompt is same as in SEMSEED.

**Solution Chain-of-Thought Prompt**

```
Solve the following coding problem. Write code in Python. Analyze the problem and the
provided test cases step-by-step and provide a solution at the end. Encolse your code in a
code block (```python...```). Please, use the starter code if it is provided (i.e. name the
function as provided in the starter code). ONLY provide the code solution at the end of
your answer. Pay attention if the task requires stdin/stdout, make sure your code runs and
reads from stdin and writes to stdout in that case. Do not add any use cases, etc.

{task}
```

Figure 7: Solution prompt by CoT baseline. The system prompt is same as in SEMSEED.

**Feedback Prompt**

```
Here are the test results for your solution.

{test_results}

Please, analyze them and try again.
```

Figure 8: SELF-DEBUGGING feedback prompt. The system prompt is same as in SEMSEED, and the solution prompt is same as used for Standard and Best-of-N.