# OpenReview forum: "Towards Algorithmic Diversity with Semantic Seed Sampling"
_ICLR.cc/2026/Conference — Submitted to ICLR 2026_

### Official Review · Reviewer_jnf2 · 2025-10-20

**Soundness:** 3
**Presentation:** 3
**Contribution:** 2
**Rating:** 4
**Confidence:** 4

**Summary:**

This paper introduces a simple idea to augment the sampling diversity of LLMs: first generate a list of seeds/hints to partition solution space, then sample solutions conditioned on a seed sampled uniformly from the generated set of seeds.

Overall, this paper presents a clear and well-motivated exploration of a simple yet practically useful idea—prompting large language models with multiple semantically distinct “seeds” to encourage diversity in generated solutions. The method is easy to implement, lightweight.
However, the paper doesn't cover existing work well and the improvements demonstrated here seem to be marginal. Results are also limited in scope (combination with one search method, with one search budget, tested on code only).
To strengthen the paper, the authors could:
(1) situate the method more explicitly within prior seeded or feature-conditioned generation approaches;
(2) extend the experiments to other domains, budgets, and search frameworks to better establish generality; and
(3) improve the semantic analyses to ensure that the measured diversity reflects genuine algorithmic differences rather than superficial lexical variation.
Despite these limitations, the work highlights an important and underexplored aspect of LLM evaluation—how to structure exploration efficiently—and could serve as a useful empirical baseline for future research on controllable diversity and structured search in large language models.

**Strengths:**

The paper tackles an important and timely problem—how to increase the diversity of generations from large language models under fixed budgets. The proposed method is refreshingly simple, easy to reproduce, and grounded in clear intuition about exploration efficiency. The presentation is well structured, the motivation is clear, and the theoretical analysis connects nicely to the empirical findings.

**Weaknesses:**

**Novelty:**
The idea is so simple it’s used by a lot of practitioners already. Showing that it works reliably remains a valuable thing of course.

**Related work:**
This paper omits several very related works. There is a series of papers using Quality-Diversity algorithms with LLMs to find a diversity of high performing solutions. In practice, this consists in evolving solutions through a genetic algorithm, while trying to cover as much as possible a space of behavioral features. Most approaches now condition generation and mutation steps on a target set of features (that would be called a seed here):
* Generating a diversity of poems: https://arxiv.org/pdf/2310.13032
* Generating a diversity of code problems (with seeds very similar to the ones here https://arxiv.org/abs/2310.10692)
* Generating a diversity of jailbreaks: https://arxiv.org/abs/2402.16822
* Generating a diversity of prompt for recursive self-improvement: https://arxiv.org/pdf/2309.16797

These are all competing methods that use  a structured space of features/seeds/hints to scaffold the generation of diverse outputs from an LLM.

Another related work that is very close to the proposed approach here is Intent Factored Generation: https://arxiv.org/pdf/2506.09659; also proposing to generate seeds with high temperature, then generate solutions conditioned on these seeds. The only difference is they generate seeds independently instead of generating a list of seeds and manually sampling from it. Unless the present paper is an iterated version of that one, this related work should be discussed.

**Control:**
There is a simple control that would be interesting to try out, it’s the generation of unrelated seeds. Maybe this works because the prompt changes a bit. Maybe we could see gains if we were to use a random wikipedia paragraph as the seed?

**Semantic analysis**
It’s difficult to interpret these analyses, because the observed differences between seed-conditioned samples might reflect superficial text effects rather than true algorithmic diversity. For example, a seed could prompt the model to mention the hint (e.g., “using recursion”) in comments or variable names, even if the actual code does not implement that strategy. In that case, the embeddings would capture lexical differences induced by the seed, while the underlying program logic remains unchanged. One way to make this more robust could be to remove all docstrings and comment before running the analysis.

What other embedding space could be used here? Maybe a binary encoding of unit tests that were passed or not?

**Methods:**
What does underlines indicate in the tables? It would be great if it indicated something like: not statistically significant from the best. So we could know from the Table if the top performing model is statistically better than others.

A lot of space could be saved by merging the two tables (with smaller font) and merging graphs. I’m not sure I understand why the results are split into two categories?

The experiments seem a bit limited in scope. Since the idea is simple, the experiments could cover different settings to really demonstrate that the idea is efficient and general.
* current results show little to no improvement (<2% increase, sometimes decrease in all but 1 task x model combination over best-of-6)
* experiments only look at combinations with one search method (best-of-n), it could also look at different search methods (eg self-consistency, or more sophisticated genetic algorithms like the ones introduced in https://arxiv.org/abs/2405.17503, https://arxiv.org/abs/2507.14172)
* experiments only look at one search budget (6)
* experiments only look at search on code problems, but could also have looked at how the approach may enhance exploration in GRPO-based training on math tasks, for instance

**Questions:**

* K means: what is the representation space and the distance metric used in the K-means? Are these high-dimensional embedding spaces? Can we rely on clusters formed in such high-dimension spaces?
* Same question for the “how much of the baseline support is covered by the seeds?” Are balls meaning anything in such high-dimensional spaces, they are so sparse?
* What is embedded: code-only? code+some description?

---

### Official Review · Reviewer_WytC · 2025-10-31

**Soundness:** 2
**Presentation:** 2
**Contribution:** 2
**Rating:** 4
**Confidence:** 3

**Summary:**

The paper proposes semantic seed sampling (SEMSEED), a training-free modification to Best-of-N for code generation: the model first emits a short list of “semantic seeds” (strategy hints like use recursion / DP / memoization), then samples a small budget of solutions per seed, followed by standard selection using public tests. The authors argue that seeds carve the solution space into disjoint strategy modes, enabling controllable, breadth-first exploration under a fixed budget. They provide bounds on pass@k for seed-conditioned sampling and a sparsity argument suggesting small per-seed budgets are near-optimal. An exploration analysis (clustering embeddings of generated solutions) reports mode separation, higher coverage than high-T sampling. Ablations show that accuracy generally improves as the number of seeds K increases (up to 6) and that higher seed temperature can help on harder suites. Limitations include imperfect separation and occasional seed neglect at high temperatures.

**Strengths:**

1. It drops into existing Best-of-N pipelines and supports expert-provided seeds.
2. Improving on pass@k bounds is good; empirical evidence of separated seed modes and sparse per-seed success rates that justify breadth-first allocation is provided.
3. Consistent gains under fixed budget across multiple models/benchmarks; largest relative lift on LiveCodeBench.

**Weaknesses:**

1. The current experiments primarily evaluate SEMSEED with relatively small seed counts (up to K = 6), whereas contemporary Best-of-N pipelines routinely use much larger candidate budgets (e.g., K ≈ 128 or more, cf. "Does Reinforcement Learning Really Incentivize Reasoning Capacity in LLMs Beyond the Base Model?"). It remains unclear whether the method preserves its relative advantage when scaled to the high-budget regime typically used in state-of-the-art code generation. A more thorough sweep over a larger K or a token-normalized comparison against higher-budget baselines would better contextualize SEMSEED’s scalability and efficiency claims.

2. The experimental coverage is largely restricted to mid-sized open models and a small set of code benchmarks. To support broader claims about semantic-mode exploration and algorithmic diversity, it would be valuable to include (i) at least one large public model and (ii) additional evaluation suites. In particular, testing outside pure program synthesis《 e.g., on math-reasoning tasks or even a lightweight ARC-style or synthesis suite, would help verify whether the proposed semantic-seed mechanism generalizes beyond code. Even a pilot study on a representative math benchmark would significantly strengthen the generalization narrative.

**Questions:**

N/A

---

### Official Review · Reviewer_NZTP · 2025-11-01

**Soundness:** 2
**Presentation:** 3
**Contribution:** 2
**Rating:** 4
**Confidence:** 4

**Summary:**

This paper introduces "Semantic Seed Sampling" (SEMSEED), a simple, training-free inference method designed to improve algorithmic diversity in LLM-based problem-solving, particularly for code generation. The core problem addressed is that standard high-temperature sampling, while prolific, often produces solutions that are only syntactically varied rather than algorithmically distinct, and can introduce errors.
SEMSEED operates in two stages:
1.	Seed Generation: It first prompts the LLM to generate a small set of $K$ semantically diverse "seeds" (i.e., solution ideas or hints) for a given task.
2.	Seed-Conditioned Sampling: It then divides the total sampling budget $N$ among these $K$ seeds (e.g., $n_i = N/K$ samples per seed). Each seed is appended to the original task prompt, and solutions are sampled from these $K$ modified prompts.
The central hypothesis is that these seeds guide the LLM to explore disjoint, semantically distinct regions of the solution space. The paper supports this claim with:
●	Theoretical Argument: It provides bounds on $pass@k$ and argues that when per-seed success rates are "sparse" (either very high or very low), a breadth-wise exploration (sampling from more seeds) is more efficient than a depth-wise one (oversampling a single idea).
●	Empirical Results: SEMSEED shows consistent performance gains over strong baselines, including Best-of-N (BoN), on code benchmarks like MBPP+ and LiveCodeBench, with relative gains up to 13.8%.
●	Mechanistic Analysis: The authors claim that seeds induce separated clusters (ARI=0.49) and cover a broader space, noting that the baseline fails to find the best seed mode in 41% of tasks.

**Strengths:**

1.	Simplicity and Practicality: SEMSEED is a training-free, "plug-and-play" inference strategy that is easy to implement and shows consistent performance gains on challenging benchmarks.
2.	Strong Empirical Results: The method outperforms strong baselines, including Best-of-N, on several benchmarks, particularly those known to have diverse algorithmic solutions (MBPP+, LiveCodeBench).
3.	Novel Theoretical Motivation: The connection made between the exploration strategy and the "sparsity" of per-seed success rates (Fig. 2) is a valuable insight. It provides a good justification for prioritizing breadth (more seeds, $K$) over depth (more samples per seed, $n_i$).
4.	Good Ablation Studies: The ablations on the number of seeds ($K$) and temperature ($T_s$, $T_g$) provide useful practical guidance and support the "breadth-over-depth" hypothesis.

**Weaknesses:**

1.	Inconclusive and Fragile Mechanistic Analysis: The paper's primary weakness is the failure to substantiate its core claim.
○	The coverage metrics are asymmetrical and partially contradictory (e.g., the 77% seed-to-baseline coverage suggests seeds miss 23% of the baseline's exploration space).
○	The lack of a baseline visualization makes it impossible to validate the claim that seeds induce new, discrete exploration modes not already present in high-temperature sampling.
○	The analysis is not robust. It hinges on a single, specific setup (K-Means, $K=5$). The paper provides no sensitivity analysis, so it is unknown if the ARI score or coverage metrics would hold with different clustering methods or different values of $K$ and $N$.
2.	Dependency on Seed Quality: The method's success is critically dependent on the LLM's ability to generate a truly diverse and high-quality set of $K$ seeds in the first step. The paper does not analyze failure modes, such as when the LLM generates $K$ semantically redundant seeds or simply bad ideas.
3.	Questionable Baseline Implementation: The very low scores for FUNCODER* raise concerns about the implementation fidelity and, by extension, the reliability of the comparisons to this SOTA method.
4.	Unmentioned Overhead: The paper claims the method is "token-efficient", comparing it to BoN. While the solution sampling stage is comparable, this ignores the non-trivial overhead of the initial seed-generation step (an extra LLM call for every task), which is not quantified or discussed.

**Questions:**

1.	Critical: Baseline Visualization: Could the authors please provide the baseline (high-temperature, T=1.0) solution embeddings, visualized using the exact same methodology (tasks, PCA, etc.) as in Figure 4? This is essential to validate the claim that SEMSEED induces a discrete cluster structure that the baseline lacks.
2.	Critical: Explaining the 77% Coverage: How do the authors reconcile the "Seed-to-baseline coverage (77%)" with the claim that the seed distribution "expands well beyond" the baseline? This metric seems to imply that SEMSEED fails to cover 23% of the solution space that the baseline can reach.
3.	Critical: Robustness of Clustering: The analysis uses K-Means with $K=5$. How sensitive are the ARI and coverage results to this specific setup? Would the conclusions still hold with different clustering algorithms (e.g., DBSCAN, hierarchical) or different values of $K$ and $N$?
4.	Budget Scaling: The experiments are fixed at $N=6$. How does the method perform with a larger total budget, (e.g., $N=32$)? Based on the sparsity theory, would $K=32, n_i=1$ (pure breadth) be optimal, or would a mixed strategy like $K=8, n_i=4$ be better (to handle "good" seeds that are not 100% successful)?
5.	Seed Generation Robustness: The paper assumes generating $K$ seeds in a single list promotes diversity. Was this compared against i.i.d. sampling (i.e., $K$ separate LLM calls) for the seeds? How does the method handle semantically redundant seeds in the generated list?

---

### Official Review · Reviewer_gUcX · 2025-11-03

**Soundness:** 3
**Presentation:** 3
**Contribution:** 1
**Rating:** 2
**Confidence:** 4

**Summary:**

The paper presents a solution to the problem of limited diversity in best-of-N sampling by breaking it up into a 2-step process---first, generating $k$ seed ideas/hints, then running best-of-N $k$ times prefixed with each of the ideas/hints.

**Strengths:**

The problem identified is sound and has the potential to broadly improve existing solutions across tasks. The paper is clearly written and well-structured, and substantial analyses have been conducted to explain the results.

**Weaknesses:**

As far as I can see, the main idea of the paper is to prompt the LLM with a set of pre-generations that can change the sampling distribution when different pre-generations are conditioned on. If that is indeed correct, I'm afraid the idea is not very novel and versions of this are now a fairly common prompting strategy when using LLMs for optimizing/searching for solutions under a fixed budget. Here are two (of likely many) examples:
1. Large Language Models to Enhance Bayesian Optimization (Liu et al.) includes a warm-starting step, commonplace in the BayesOpt literature, executed via LLMs in this work to generate a set of diverse initial solutions to build further on. See also trust-region methods.
2. Open-ended Scientific Discovery via Bayesian Surprise (Agarwal et al.) uses tree search methods (which in themselves are not novel to this paper), where sampling at the root node is the equivalent of the first step in the proposed method, and sampling by conditioning on the resultant nodes is the second step.
3. As is also mentioned in the paper, CodeChain and FunCoder use the same idea at the component level. But simply not operating at the component level is insufficient to qualify as having added novelty, in my opinion.

My review may have been different if there was some novelty in how the seeds were generated that worked better than other techniques. However, unless I have misunderstood the key novelty in this paper, I do not think there is a case for a publication in a main conference.

**Questions:**

1. Are the main results reported on a single run, or averaged across multiple runs? If the latter, could you please report the standard deviations?
2. Is the proposed method intended for low-budget scenarios (e.g., 6) or for much larger budgets (e.g., 500) as well?

---

### Meta-Review · Area_Chair_ewwx · 2026-01-06

**Summary:**

The paper proposes semantic seed sampling (SEMSEED), a training-free modification to Best-of-N for code generation. All of the reviewers agree that the paper lacks novelty, only have small scales of random seed experiments, and omits several very related works.

Specifically, reviewers raise concerns about:
1) the core idea of prompting an LLM with pre-generated seeds to shift the sampling distribution is already a common strategy in the literature;
2) the analysis intended to explain why the method works was found to be unconvincing and lacking rigor;
3) the empirical evidence was characterized as narrow and the performance gains as marginal.

**Reviewer Concerns:**

All concerns remain unsolved as no rebuttal is posted.

**Reviewer Scores:**

The reviewers will remain their score as no rebuttal has been posted.

---

### Decision · Program_Chairs · 2026-01-26

Reject